# The Adenine/Thymine Deleterious Selection Model for GC Content Evolution at the Third Codon Position of the Histone Genes in *Drosophila*

**DOI:** 10.3390/genes12050721

**Published:** 2021-05-12

**Authors:** Yoshinori Matsuo

**Affiliations:** Division of Science and Technology, Tokushima University, 2-1 Minamijosanjima-cho, Tokushima 770-8506, Japan; matsuo.yoshinori@tokushima-u.ac.jp; Tel.: +81-88-656-7270

**Keywords:** histone gene, GC content, *Drosophila*, codon usage, size effect

## Abstract

The evolution of the GC (guanine cytosine) content at the third codon position of the histone genes (*H1*, *H2A*, *H2B*, *H3*, *H4*, *H2AvD*, *H3.3A*, *H3.3B*, and *H4r*) in 12 or more *Drosophila* species is reviewed. For explaining the evolution of the GC content at the third codon position of the genes, a model assuming selection with a deleterious effect for adenine/thymine and a size effect is presented. The applicability of the model to whole-genome genes is also discussed.

## 1. Introduction

Histones are basic proteins that package and arrange DNA into nucleosomes [1,2,3,4]. There are two major types of histones: a replication-dependent (canonical) type and a replication-independent (replacement) type [5]. In addition to these, centromeric proteins [6,7,8] and histone-like proteins [9] also exist.

In *Drosophila*, five replication-dependent (canonical) histones are known [10,11]: H2A, H2B, H3, and H4, which are core histones that organize the nucleosome core by forming an octamer comprising two copies of each protein, and H1, which is a linker protein that binds to each nucleosome core [1,2,3,4]. As for replication-independent (replacement) histones, four kinds are currently known in *Drosophila*: H2AvD, H3.3A, H3.3B, and H4r [12,13,14,15]. In addition to histone modification [16,17,18,19,20,21,22], the replacement of histones by a different histone type causes chromatin remodeling [23,24,25]. Nucleosome remodeling is involved in many important biological processes, such as cell division, differentiation, gene expression, and replication [26,27,28]. Therefore, histone modification and replacement are mechanisms that can lead to epigenetic changes [21,29,30]. In *Drosophila*, the histone genes for the canonical type of histones are clustered in a repetitive unit, and in *Drosophila melanogaster*, the unit repeats about 110 times [10,31]. In contrast, the histone genes for the replacement type of histones are found as single genes or with only a few copies per genome, and they contain a few introns [12,13,14,15]. For the detailed structure of the histone genes in *Drosophila*, please refer to another review article [32]. The mode of molecular evolution of a multigene family, compared to a single gene, can be studied by analyzing histone genes [33].

The usage of codons in protein-coding genes is not uniform among synonymous codons and is biased in many species [34,35]. The mechanism of codon bias has been discussed for decades, and candidate factors include mutation bias, natural selection, and genetic drift [36,37,38,39,40,41,42,43,44]. Unequal usage of codons occurs when the rate of mutations due to nucleotide substitutions is biased or when selection pressure is exerted differently between synonymous codons. Fitness differences among synonymous codons may be present due to differences in the efficiency or speed of translation [45,46]. However, the selection pressure on codons, if any, would be comparatively weaker than that on amino acid sequences; therefore, the codon usage can be influenced by population size [32,39,47,48,49,50,51,52,53]. Since the largest difference in codon usage is observed in the nucleotide at the third codon position of genes, the guanine–cytosine (GC) content at the third codon position is strongly related to codon usage bias. In *Drosophila*, the higher the GC content at the third codon position is, the stronger the bias of codons [37,40,54]. Moreover, regarding the relationship with the evolutionary rate, the stronger the bias of codons is, the slower the evolutionary rate [55].

In *Drosophila saltans*, the low GC content of the *Xdh* and *Adh* genes was explained by fluctuating mutation bias [56,57]. However, it may also be explained by changes in selection [32,38,50,51,52,53]. Although many *Drosophila* species have been analyzed for their histone genes [31,49,58,59,60], no changes in the rate of mutations were observed among the species in our analysis [49]. Here, the evolution of the GC content at the third codon position of histone genes in *Drosophila* is reviewed, and a model that can best explain the evolution of the GC content at the third codon position in *Drosophila* is presented.

## 2. Evolution of the GC Content at the Third Codon Position of the Histone Genes in *Drosophila*

The GC content at the third codon position of the histone genes in 12 *Drosophila* species is shown in Figure 1. Parts of histone genes data have been published from our laboratory [31,49,51,52,58,59,60]. The rest is obtained from FlyBase (http://flybase.org, accessed on 2017–2019) [61]. Several characteristic points on the evolution of the GC content at the third codon position of histone genes in *Drosophila* are summarized below.

### 2.1. Disparity in the GC Content at the Third Codon Position among the Genes

In many *Drosophila* species, the codon usage of the genes was uneven and varied from gene to gene [36,40,50]. Therefore, the GC content at the third codon position differed between the genes. Although the reason remains unclear, codon bias was found to be related to the level of gene expression, which also varied from gene to gene. The positive relationship found between codon bias and the level of gene expression most likely resulted from the difference in translation efficiency [45,46]. Among the canonical histone genes, *H2B* showed the highest GC content at the third codon position, while *H1* showed the lowest GC content at the third codon position [62]. H1, a linker protein, is expressed at approximately half of the level of the other four canonical histones. This is likely the reason why the GC content at the third codon position of *H1* is not as high as those of the core histone genes.

### 2.2. Disparity in the GC Content at the Third Codon Position between the Genes of the Canonical and Replacement Types of Histones

A comparison of the average GC content at the third codon position of genes in 12 common species revealed a higher GC content at the third codon position in the genes of the replacement type of histones than in those of the canonical type of histones [62,63,64,65]. Analysis of codon bias in the histone genes demonstrated that the difference was caused not by an obvious codon bias in a specific amino acid but by a general tendency that was observed for many codons [62]. Differences in functional differentiation or translation efficiency may be the cause of the differences in GC content at the third codon position between the histone types.

### 2.3. Disparity in GC Content at the Third Codon Position of the Genes among the Different Species

Although variability in the GC content among the genes within a species has been previously noted [36,40,50,51], variability has also been observed between different species [40,51,62]. For example, among 12 *Drosophila* species, the GC content at the third codon position of many genes in *Drosophila willistoni* was relatively lower than in the other 11 species [39,62]. Furthermore, when the GC content at the third codon position of corresponding genes was compared between the *Drosophila* species, nearly parallel differences, similar patterns of ups and downs, were observed for most comparisons (Figure 2). A lower GC content at the third codon position was also observed in the genes of *Drosophila* species other than these 12 species, such as in *Drosophila hydei* and *Drosophila americana* (Figure 1).

### 2.4. Mode of the Evolution of GC Content at the Third Codon Position According to Phylogeny

The differences in GC content at the third codon position according to the *Drosophila* phylogeny were unexpected and lacked consistency with evolution [33,62]. The GC contents at the third codon position of closely related species showed similar values, but those in distantly related species did not always show larger differences. Unlike the case for nucleotide and amino acid substitutions, the relationship between differences in GC content at the third codon position and the evolutionary distances between species is not co-linear. The differences in GC content at the third codon position are independent of phylogenetic distance.

## 3. Models for the Evolution of GC Content

### 3.1. Mutation Bias

Candidate genetic factors that can explain the evolution of GC content including selection, genetic drift, and mutation [36,41,57]. The patterns and rates of mutations are hard to measure directly by experiments, but they can be estimated by several indirect methods, for example, by analyzing the bases, or GC content, in regions that are free from selection [40,41]; such regions include long introns, intergenic spacers, and regions of transposons without functional constraints. The GC content of such regions is most likely determined by mutation bias alone. Therefore, if the effect of mutations in these regions is stationary, the patterns and rates of mutations can be estimated by the content of each base in these regions. In most cases, these regions are observed to be AT rich, meaning that the mutation is biased for adenine/thymine (A/T) [40,41]. For example, in the *D. melanogaster* histone gene cluster, the GC content of the longest spacer between the *H1–H3* genes was observed to be the lowest at 30% [31,32,62]. Therefore, a mutation in *D. melanogaster* must be biased, at least a little, for A/T.

It is also possible to test whether or not the mutation bias varies among species by comparing the GC content of these regions. If the GC content in a broad range of species is distributed within a narrow range, then the mutation pattern and rate must be stable among those species. For example, the GC content of the *H1–H3* spacer region in *Drosophila* is approximately 30% in a broad range of species [51,62]; thus, the mutation bias in *Drosophila* must be stable, and it is unlikely that a variation in GC content at the third codon position among the species is the result of a fluctuating mutation bias among the species. A low GC content can be easily explained by mutation bias because the mutation was biased for A/T. However, a high GC content cannot be explained merely by mutation bias; to explain a high GC content, other genetic factors, such as selection and drift, need to be considered.

### 3.2. Deleterious Selection for A/T

Since the GC content was lowest in regions free from selection and was most likely determined by mutation bias alone, to explain the observed codon usage and GC content at the third codon position described above, selection resulting in G/C increases or A/T decreases was assumed to have been involved. Here, an evolutionary observation has to be taken into account for considering the assumptions of our model. It is a fact that most new mutations biased for A/T are observed to be either neutral or deleterious [66,67]. This can again be explained by the selection that is deleterious for A/T. Therefore, these observations on the GC content at the third codon position can be most easily explained by assuming that selection for A/T is mostly, but not completely, deleterious. In regions with a high GC content, there is selection acting for the removal of A/T, and in regions with a low GC content, the selection is weak or not acting at all. However, the selection coefficient for A/T nucleotides, which varies from codon to codon in genes, must be much smaller than that of the amino acid sequence. Therefore, the overall selection for codon bias in a gene seems to be a “gene effect.” For example, the GC content at the third codon position of the *H1* gene is lower than those of the other histone genes. Another example is the difference in GC content at the third codon position among the different histone types, which must be due to the functional differentiation between the histone types, such as the efficiency and rate of translation [62].

### 3.3. Effect of the Population Size

The selection for A/T at the third codon position was weak enough for it to be affected by the population size. In the nearly neutral model of molecular evolution, the selection is more effective on larger populations but not as effective on smaller populations [47,48]. The deleterious selection for A/T in a large population would thus be expected to be effective, resulting in a high GC content at the third codon position and high codon bias. In contrast, in a small population size, the deleterious selection for A/T would not be expected to be effective, and the GC content at the third codon position would remain low with a low codon bias. Another point to be noted here is that the whole genome is affected by the size effect, that is, all genes in the same genome simultaneously experience the same size effect. Furthermore, in the past, the genomes of species must have experienced repeated changes due to changes in the population size. Therefore, the increases and decreases in GC content are expected to be linked for all genes in the same genome. The accumulation of past changes in the genomic genes must be a “species effect.” In *D. willistoni* and several other *Drosophila* species, the GC content of the genomic genes was considerably lower in comparison to that in other *Drosophila* species [39,41,51,62]. In the species with a low GC content, none or a weaker selection must have been acting. Therefore, in those species or their ancestors, a decrease in the population size must have occurred [51].

## 4. Generality of the Model

Although the evolution of the GC content at the third codon position of the histone genes in *Drosophila* can be most easily explained by the deleterious selection for the A/T model, data for more genes and from more species should be analyzed in detail to confirm whether this model can be applied to other genes in the genome. More than 6000 genomic genes from each of 12 *Drosophila* species were analyzed for codon bias by Vicario et al. [39]. The following facts used for constructing the above model are applicable to histone genes, as well as many other genes in the genome: (1) there is variability in the GC content at the third codon position or codon bias of the genes in a species that is hard to explain by mutation bias alone; (2) the GC content was lowest in the regions with weak or no selection, such as the introns and spacers, in a broad range of species, and the level was similar between the different species [32,39,41,52,60,62]; (3) in *D. willistoni*, the GC content at the third codon position was lower in most genes, and the codon bias was relaxed when compared to the other species [39,41]; (4) the GC content at the third codon position of corresponding genes among *Drosophila* species tended to show a nearly parallel difference for each comparison (Figure 2). Therefore, it is “unlikely” that the deleterious selection for the A/T model is applicable only to the histone genes. Although many more genes and species should be analyzed, the results from the analysis of histone genes appear to be applicable to all genes.

## 5. Conclusions

The evolution of the GC content at the third codon position of histone genes in *Drosophila* was reviewed. The model that can best explain the observed data is the deleterious selection for the A/T model with the population size effect.

## Figures and Tables

**Figure 1 genes-12-00721-f001:**
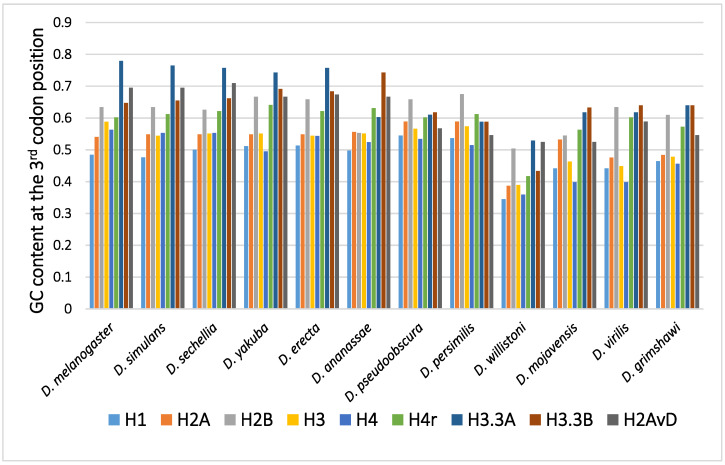
The GC content at the third codon position of the histone genes in *Drosophila*. The data grouped according to the *Drosophila* species.

**Figure 2 genes-12-00721-f002:**
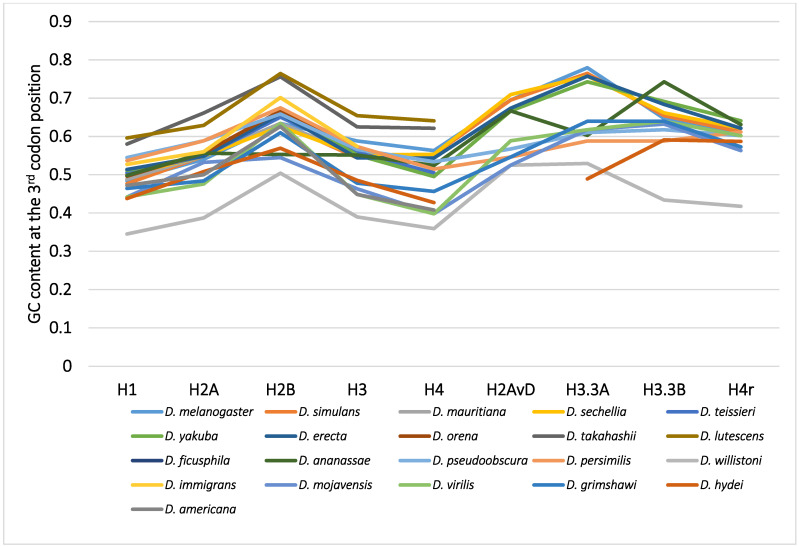
The GC content at the third codon position of the nine histone genes in *Drosophila*. The points from the same species were connected by lines to show the trend for each species.

## Data Availability

No report for available data.

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
