# Peer review of "The Adenine/Thymine Deleterious Selection Model for GC Content Evolution at the Third Codon Position of the Histone Genes in Drosophila"

_genes, 2021, doi:10.3390/genes12050721_

Round 1

Reviewer 1 Report

In this article, the author briefly summarizes the current knowledge about the evolution of the GC content in the genome, mainly focusing at the content of the 3rd codon position. In this context, the author discusses and examines some data previously published by his lab and others about the histone family genes in Drosophila species.

In general, I find the manuscript quite clear and logically structured, and the two figures are clear and support the logic of the main text. However, I would recommend addressing some aspects that might improve the manuscript:

- In the way the introduction is currently structed, it is confusing which is the actual aim of the review: the study of the evolution of histone genes in Drosophila or the evolution of the GC content at the third codon. My understanding is that histone genes in Drosophila are just used as an example to explore the evolution of the GC content at the third codon. To clarify this, I would recommend to start the review describing the usage and evolution of codons (paragraph currently starting at line 36), and then introduce histones as a conserved gene family that it is useful to explore this, highlighting the advantages of using this gene family and not others. Otherwise, if the author actually aims to discuss about the histones for any particular reason, that should be described.

In line with the above, I would find quite helpful to add some information to link paragraphs 1 and 2 (lines 35 and 36), whichever goes first. Introducing for example, why it is interesting to study the usage of codons in histone genes, or why histones genes are a good model to explore questions related with the usage of codons.

- Sometimes (for example in sections 2.1 or 3.1), I find difficult to follow the logic of comparing the GC content at the 3rd codon position with the GC content in general. Or in other words, why data about these two types of CG content can be mixed and directly compared to support conclusions? (Beyond the fact that more GC content probabilistically should reflect more GC content at the 3rd codon). Section 2.2 is an example where the title refers to the GC content at the 3rd codon, but where only data related to general CG content is described. I would find quite helpful some clarification about this, at least at the introduction of the article. In the current version, only information about the relevance of the content at the 3rd codon position in provided (lines 45-48).

Reviewer 2 Report

The  large number of Drosophila species which are separated by many millions of years (they might provide the time scale of their evolutionary comparison) provide an excellent group of species for this study. In addition to the 3rd bases, both the arginine and leucine codons, have the potential to start with either a C or an A/T, and they might mention whether the CTN leucine codons are preferred over TTA/G, and whether the CGN arginine codons are preferred over AGA/G, since that is another place where single nt changes could occur that changed the GC contribution.

A problem with the paper is that there are not sufficient specific conclusions drawn.  They conclude the H1 coding region is under the least selection since that gene is expressed less than the other histone genes in the cluster, consistent with the possibility that it does not need to be translated at as high a rate as the other genes. That interpretation makes sense. 

However they make no mention of the fact that the GC content of the other four genes in the cluster is lower than the single-copy histone genes.  Certainly this looks like as striking a difference as the difference between the H1 gene and the other genes in the cluster, and should be discussed.  Perhaps the fact that the clustered genes are present in multiple copies reduces the need to have efficient translation of the mRNA, since the number of genes are in clear excess over what is necessary.

 Since the genes with single copies have the highest G/C content that seems consistent with the need to express these proteins effectively. 

The other thing that stands out in that the H2b gene in the cluster is different than the other genes in the cluster, tending to have a higher GC content closer to the single copy genes.  This needs at least to be mentioned, not clear that this .        

One assumption of this work is that there is no role for the spacer between the H3 gene and H1 gene, which is the longest spacer in the repeat unit in all Drosophila species, and really the only one that has no role in some known aspect of expression of the mRNA.  The A/T content of this region is similar to that of the regions in ref. 38 that are not underselection, but which are a much larger group of sequences.

I would add something about how conserved (or what variation there is in length) in that spacer.  Single mutations in this region obviously does not have a constraint on coding potential, since it is not expressed.  However, it could play a role in some aspect of organizing the repeat unit, and if the length has been conserved, the length could be a clue to that..

2. Ref. 38 is in BMC Ecology and Evolution.  The journal was omitted in the bibliography.
